# Study of Physico-Chemical Properties of Dough and Wood Oven-Baked Pizza Base: The Effect of Leavening Time

**DOI:** 10.3390/foods12071407

**Published:** 2023-03-26

**Authors:** Clelia Covino, Angela Sorrentino, Prospero Di Pierro, Paolo Masi

**Affiliations:** 1Department of Agricultural Sciences, University of Naples Federico II, Via Università 100, 80055 Portici, Italy; 2Centre for Food Innovation and Development of the Food Industry, University of Naples Federico II, Via Università 133, 80055 Portici, Italy

**Keywords:** elastic modulus, stress-relaxation, starch gelatinization, digestible starch, gluten

## Abstract

The research objective was to investigate the morpho-rheological, chemical, and structural changes of dough and Neapolitan pizza TSG as the leavening time varies and to evaluate their effects on the digestibility of starch and on the formation of acrylamide during baking. Pizza dough leavening was monitored for 48 h at 22 °C/80% RH, and the analyses were conducted at selected leavening times (0, 4, 8, 16, 24, and 48 h). It was observed that in 30 h the volume tripled and the viscoelastic dough relaxed in the first 4 h, as evidenced by the lower value of the relaxation percentage “*a*” and the higher rate of decay “*b*”, associated with a high value of the compression work, indicating the presence of a very strong gluten mesh. In the following hours, the dough lost elasticity, and in fact, the G’ modulus decreased due to the weakening of the weak interactions between the gluten proteins and the starch. This suggests that a long leavening improved the extensibility of the pizza disc, facilitating the action of the pizza maker. Thermal (TGA and DSC) and morphological (SEM) analyses evidenced the highest water removal rate from the dough, a wider starch gelatinization temperature range, a ∆H of 0.975 ± 0.013 J/g, and a more open and weak gluten structure in dough balls leavened for 16 h. As the leavening time increased, both dough and pizza base samples showed an increase in reducing sugars and free amino groups, while the rapidly digestible starch decreased in the dough following the metabolism of the yeasts and increased in the pizza base due to the starch gelatinization that occurs during baking, which makes it much more susceptible to α-amylase. Finally, the levels of acrylamide remained at the same values despite the higher availability of reducing sugars and its precursors during leavening.

## 1. Introduction

Despite being a widely consumed product, Neapolitan pizza has not been the subject of exhaustive scientific studies. In 2010, the European Union Commission passed a regulation governing the production of Neapolitan pizza called TSG (Traditional Specialty Guaranteed) [1], and in the prelude to that document, the appearance of Neapolitan pizza was traced back to a historical period between 1715 and 1725. Neapolitan pizza spinning is considered an art and, on 7 December 2017, received recognition by UNESCO (the United Nations Educational, Scientific, and Cultural Organization) as an “Intangible Cultural Heritage of Humanity”. This gave the art of making pizza global significance [2]. Perhaps thanks to this important recognition, since 2010, research on pizza has undergone a significant boost. Thus, in recent years some characterization studies of Neapolitan pizza have been published with the intention of emphasizing its nutraceutical properties [3], evaluating the influence of ingredient quality on the physico-chemical and sensorial properties [4,5], and even replacing the salt with sea water [6]. Moreover, the pizza base was fortified to improve the antioxidant properties [7,8] and enzymatically treated with asparaginase to explore the possibility of reducing the formation of process contaminants such as acrylamide [9]. Further studies have been directed to interesting aspects in the “world of pizza,” such as the characterization of cooking methods in a wood-fired or electric oven [10,11] up to the point of even hypothesizing 3D printing for both a classic and a gluten-free pizza [12]. However, in the panorama of scientific literature on Neapolitan pizza TSG, there is a lack of a complete 360-degree study that monitors the evolution of pizza from the leavening of the dough to baking in a wood oven, analyzing rheological and structural parameters together with the thermal and biochemical properties without neglecting aspects related to health and food safety.

Pizza dough is a complex, viscoelastic material, and its composition plays an important role in the dough’s processability, the gas holding capacity during fermentation, and the baking performance [13]. The formation of the gluten network is also fundamental during mixing, in which starch and water are included, and can affect subsequently the development of the dough during leavening.

Ripening is a set of enzymatic processes that progressively break down more complex structures, like proteins and starches, into simpler elements, namely peptides/amino acids, and fermentable sugars for yeasts. Brewer’s yeast is the main leavening agent in the pizza-making process. During leavening, yeast (*Saccaromyces cerevisiae*) metabolism switches from respiration to fermentation, converting fermentable sugars into ethanol and carbon dioxide. During fermentation, yeast cells produce CO_2_, which partly dissolves in the aqueous phase of the dough and forms weakly ionizable carbonic acid that slightly lowers the pH of the dough. When this phase is exhausted, the carbon dioxide subsequently produced may vaporize into the environment or pass into the air as microbubbles formed within the dough during the mixing phase. As the carbon dioxide passes into the bubbles, they expand due to the increase in pressure, generating an overall increase in the volume of the dough and the formation of gaseous alveoli [14].

The baking process of pizza is accompanied by the evaporation of water, the denaturation of protein, and the gelatinization of starch. During the gelatinization process, water breaks down the crystallinity of the starch, the granules swell, amylose diffuses out of the granules, leaving most of the amylopectin behind, and the granules eventually collapse and are held in an amylose matrix as part of a gel network. High temperatures denature proteins, and as a result of denaturation, proteins can undergo extensive cross-linking, particularly through the formation of disulfide bonds, thus forming a continuous protein network. When proteins and starch are in contact with each other, stable complexes can develop by forming a protein/starch matrix, where hydrogen and covalent bonds as well as charge-charge interactions can be found [15].

Rheological analysis is commonly used to assess the properties of dough and gain insights into the functions of ingredients and dough structure [16]. To this aim, a stress-relaxation test, which involves applying a strain to the dough, could be used to see how the dough relaxes over time, while the dynamic oscillatory rheometer can be useful to describe the molecular structure formation of starch and gluten during baking and cooling. In cereal science, scanning electron microscopy (SEM) could be used to show the morphology of the dough structure at different leavening times [17].

The purpose of our study was to investigate the change in the structure of pizza dough during leavening and baking through physico-chemical analyses and to correlate the dough structure at selected leavening times (0, 4, 8, 16, 24, and 48 h) with the starch digestibility of pizza bases baked in the wood oven. For each considered leavening time, physical properties such as rheological, thermal, and morphological were studied, and the biochemical aspect was evaluated by determining reducing sugars, free amino groups, and *in vitro* starch digestibility, which may be associated with the glycemic response of the food. Lastly, in terms of food safety, the quantity of acrylamide that forms in the pizza base following baking in a wood oven was also determined.

## 2. Materials and Methods

### 2.1. Chemicals and Pizza Dough Ingredients

Chemical reagents were of analytical grade and were obtained from standard chemical companies. The Digestible and Resistant Starch Assay Kit was purchased from Megazyme Ltd (Wicklow, Ireland). The soft wheat flour used for the experiments was the refined commercial type “00” (Caputo rossa) kindly provided by Mulino Caputo (Antimo Caputo, Srl, Naples, Italy), whose proximal composition shown on the label was: 70% carbohydrate, 13% protein, 12% moisture, 3% fiber, 1.5% lipids, and 0.5% ash. Fresh brewer’s yeast (Lievital, Italy) and salt (common fine table food-grade sodium chloride) were purchased at a local supermarket (Portici, Italy).

### 2.2. Dough and Pizza Making

Pizza dough was prepared following the Commission Regulation (EU) n. 97/2010 to obtain a product recognized as “Neapolitan Pizza TSG” and kneaded with the traditional ingredients (60.35% flour, 37.72% water, 1.88% salt, and 0.04% yeast). After resting for 20 min at room temperature, the dough was divided into 250 g balls, which were leavened at 22 °C and 80% relative humidity (RH) for several leavening times (0, 4, 8, 16, 24, and 48 h), and then rolled and baked for 60 s in a wood oven at about 485 ± 30 °C [9].

### 2.3. Image Analysis of Leavening Kinetics

The leavening of the pizza dough balls (250 g) was monitored for 48 h by using a digital camera (Go Pro Hero 5) positioned to always frame the dough at the same distance for the entire leavening period (Appendix A) and scheduled to take pictures every hour. The obtained images were used for the measurement of the height (*h*) and width (*l*) of the pizza dough through an image analysis software (Image J, ver. 1.52t, National Institutes of Health, Bethesda, MD, USA). The geometry of the dough ball was assimilated to that of an oblate semi-spheroid and thus the volume (*V*) was determined using the following formula:(1)$$V=\frac{\;\frac{3}{4}\;\pi {\left(\frac{l}{2}\right)}^{2}h\;}{2}$$

The calculated volume at each hour of leavening was expressed as the V_t_/V_0_ index, where V_t_ is the volume measured at time t and V_0_ is the initial volume of the dough, and plotted as a function of leavening time.

### 2.4. Stress-Relaxation Test

Pizza dough at various leavening times was subjected to the stress-relaxation test using a dynamometer Instron (mod. 5900R, Norwood, MA, USA) equipped with two plates with Ø 30 cm (Appendix A), in which the pizza dough was compressed until a final thickness of 2 mm was reached; the plates were specially designed to simulate the dough rolling by the pizza maker. The tests were conducted in air at 23 ± 2 °C by applying a compressive force at a constant speed of 50 mm/min with a 3 kN load cell. Once the set thickness was reached, the test was continued without applying any other compression in order to analyze the relaxation stage of the dough. Force versus time curves (Appendix A) were recorded with Bluehill software (ver. 3, Instron, Norwood, MA, USA), which was also used for the calculation of the area under the force curve, which corresponds to the work (J) performed by the crosshead to deform the dough to the set thickness.

The relaxation graphs were obtained by plotting *F_t_*/*F*_0_ as a function of time (Appendix A), where *F_t_* is the force measured at time t and *F*_0_ is the force measured at the starting point of the relaxation test, immediately after the end of the compression stage. The extent of the relaxation percentage (“*a*”, %) and the decay rate (“*b*”, N s^−1^) during relaxation were calculated through the mathematical processing of the raw data following Equation (2) [18]:(2)$$\frac{tF_{0}}{F_{0}-F_{t}}=\frac{1}{ab}+\frac{t}{a}$$

For each sample, eight specimens were analyzed, and at the end of the test, the diameter of the stretched dough was measured for all analyzed samples in four replicates. All values were expressed as mean ± SD (standard deviation).

### 2.5. Elastic Modulus Determination

The samples at various rise times (0–48 h) were analyzed for elastic modulus determination by performing the dynamic-mechanical test using the rotational rheometer (Haake™Mars™, Thermo Fisher Scientific Inc., Waltham, MA, USA) with controlled stress and 80 mm flat and parallel plate geometry. The distance between the two plates (gap) was set at 2 mm, and three replicates were performed for each sample. Eight grams of freeze-dried and ground dough samples (grain size < 0.5 mm) were weighed, and 4.8 mL of H_2_O_d_ were added to the powder to obtain a rehydration of 60%. All samples were kneaded for 2 min and placed on the parallel plate of the rheometer, forming a dough disc covering the whole surface. The upper plate was brought closer to the lower plate at a distance of 2 mm to ensure that the sample adhered well to the surface of the plates, and excess sample was removed with a spatula, after which the edges of the sample were sealed with vaseline ointment to prevent drying of the dough during the test. The elastic component of the modulus was measured using the sweep temperature test with an oscillation frequency of 1 Hz and a strain of 0.1%. The specimens were heated from 30 to 90 °C and then cooled from 90 to 30 °C with a variation of 10 °C/min [19]. The heating rate was designed in a way to follow the thermal history of pizza dough during baking.

### 2.6. Thermal Analysis

The thermal properties of dough at different leavening times were measured by differential scanning calorimetry (DSC) and thermogravimetric analysis (TGA). DSC (Q200, TA Instruments, New Castle, DE, USA) was used under a nitrogen atmosphere (50 mL/min). Different fresh dough samples (ca. 40 mg) were placed in DSC aluminum pans and sealed; the sample and empty pans were put in the DSC testing cell and heated in the temperature range of 20–100 °C with a heating rate of 10 °C/min [20,21]. Initial (T_i_), gelatinization (T_g_), and end peak (T_e_) temperatures and gelatinization enthalpy (ΔH) were calculated from DSC thermograms through the TA Universal Analysis 2000 software (ver. 5.5, TA Instruments, New Castle, DE, USA). Results were analyzed from three replicates.

For the TGA analysis, fresh dough samples (ca. 60 mg) were placed in oxidized aluminum pans and loaded into a thermogravimetric instrument (TGA7, Perkin Elmer Inc., Waltham, MA, USA); then, a ramp temperature in the range of 30–550 °C with a heating rate of 10 °C/min was applied under a nitrogen atmosphere (50 mL/min). The thermogram curves were derived to obtain the graphs of the weight loss rate (%/°C), from which the weight loss (%) in the interval between 100 and 200 °C was calculated, which corresponds to the area of the 1^st^ peak. Data processing was conducted with the dedicated Pyris Instrument Managing Software (ver.11, Perkin Elmer Inc., Waltham, MA, USA). Each measurement was performed in triplicate and expressed as the mean ± SD.

### 2.7. Reducing Sugars, Free Amino Groups, and Acrylamide Analysis

For each leavening time, freeze-dried and ground samples of dough and a wood oven-baked pizza base were weighed, and H_2_O_d_ (1:10) was added to extract soluble substances. Next, the samples were vortexed for 3 min at room temperature (20–25 °C) and then sonicated for 15 min in an ultrasonic bath at room temperature at 240 Watt (cycle 0.5). Subsequently, samples were centrifuged (Microfuge 18 Centrifuge, Beckman Coulter) at 9000× *g* for 10 min at 25 °C to separate the soluble fraction from the insoluble fraction. The supernatants were stored at −20 °C until subsequent quantitative reducing sugars and free amino groups were determined.

The 2,4-dinitrosalicylic acid (DNS) assay for reducing sugars such as glucose was used, according to the method of Miller [22]. Determination of free amino groups was performed using the o-phthaldialdehyde (OPA) method [23]. 20 μL of supernatant were used in both reaction assays. Absorbance was measured through a spectrophotometer UV/VIS (V-730, Jasco, Easton, MD, USA) at 540 nm and 340 nm for the DNS and OPA assays, respectively. Four measurements were made for each sample analyzed. A calibration curve was built for each assay using glucose (0–0.16 mg/mL) and leucine (20–300 mg/L) as standards. Reducing sugars were expressed as g of glucose equivalent/100 g of sample on a dry weight (dw) basis (g GE/100 g_dw_), while free amino groups were referred to as g of leucine equivalent/100 g of sample on a dry weight basis (g LE/100 g_dw_).

Freeze-dried and ground samples of wood oven-baked pizza bases were subjected to AA determination. The details of the acrylamide (AA) extraction and determination protocols were previously reported by Covino et al. [9]. The recovery of AA extraction for the samples analyzed in this study was in the range of 86–92%. 

### 2.8. In Vitro Digestible and Resistant Starch

The determination of digestible and resistant starch in dough and pizza samples was carried out following AOAC method 2017.16 (Digestible and Resistant Starch Assay Kit, Megazyme, Wicklow, Ireland). For the determination of digestible and resistant starch, the procedure provided by the manufacturer’s instructions for the kit was slavishly performed on freeze-dried samples of 0.5 g. Rapidly digestible starch (RDS, after 20 min), slowly digestible starch (SDS, after 120 min), and resistant starch (RS, not digested after 240 min) percents (w/w) were determined using the following formulas:DS (%) = ∆A × F × EV/W × 0.0189(3)
RS (%) = ∆A × F × EV/W × FV × 0.000225(4)

RDS, SDS, and RS values were expressed with respect to dry matter in order to avoid the influence of water content on the results of the determined starch fractions. The starch digestion rate index (SDRI) was calculated as RDS divided by Total Starch (TS = RDS + SDS + RS) and represents an indicator of *in vitro* starch digestibility. The rapidly available glucose index (RAG) was determined as a predictor of the potential glycaemic response derived from the ingestion of these food items [24].

### 2.9. SEM of Dough and Pizza Samples

SEM analyses were conducted for the evaluation of dough samples at selected leavening times. Lyophilized dough slices [25] were placed on specimen holders and coated with gold by means of DC sputtering (Sputter and Carbon Coater Agar Scientific B7340). The microstructure of samples was observed at a magnification of 2000× with a LEO EVO 40 SEM (Zeiss, Oberkochen, Germany) with a 10 kV acceleration voltage. Representative micrographs from all the samples were selected. The microstructure of lyophilized wood oven pizza base samples was observed at a magnification of 2000× by a Phenom XL SEM (Thermo Scientific, Inc., Waltham, MA, USA) equipped with an ion sputter coater.

### 2.10. Statistical Analysis

All experiments and analytical measurements were performed in triplicates, and data were expressed as mean ± SD. The means of each parameter were analyzed by analysis of variance (ANOVA) using Post Hoc Tukey’s test. Statistical analyses were performed using XLSTAT software (version 2014.5.03). Differences between treatments at the 5% level (*p* < 0.05) were considered significant.

## 3. Results and Discussion

### 3.1. Dough Volume during Leavening

The leavening kinetics of pizza dough were monitored for 48 h using digital image acquisition, and the volume of leavened dough was obtained through an image analysis software. The variation of the volume index (V_t_/V_0_) over time shows a typical sigmoid kinetic curve, characterized by three phases: the first phase, in which the curve has a very slight positive slope and the V_t_/V_0_ just reaches 1.3 after 8 h of leavening; the second phase, which is the real growing phase, in which the volume increases in a logarithmic way until about the 30th h, by tripling its initial value; and the third phase, which corresponds to the stationary phase, when the volume index runs along the asymptote of the described curve (Figure 1A). The increase in volume has been linked to the production of CO_2_ during leavening. At the mixing of the ingredients, air bubbles were introduced into the dough, which were then replaced during the leavening by CO_2_ produced by the fermentation of yeast (*S. cerevisiae*).

During fermentation, the sugars in the dough were converted by the yeast mainly into CO_2_ and ethanol. The amount of CO_2_ produced depends on the fermentable substrates in the dough. The content of free fermentable sugars in wheat flour was too low to support optimal gassing power by yeast cells. The sugars that were consumed during fermentation were generated by the enzymatic hydrolysis of damaged starch [26]. The size and density of the bubbles can change the texture and sensory properties of the finished product [27]. It is worth noting that in the early stage of leavening, a change in the shape of the dough ball was observed, which was more flattened by gravity and enlarged at the base, but the overall volume remained constant (Figure 1B, images at 4 and 8 h). At later times, as the CO_2_ produced by the yeast increased in the growing phase, the volume increased (Figure 1B, images at 16 and 24 h) until the stationary phase. In fact, during this time, the cell structure definitely changed as the size of the bubbles increased, which resulted in stretching of the dough matrix due to the bubble growth (Figure 1B, image inserts at 16 and 24 h) [28,29]. In the stationary phase, the total volume does not change, but the dough undergoes flattening phenomena that are very evident at the late stage (Figure 1B, image 48 h). This behavior is typical of viscoelastic materials that relax during leavening, due to destructuring phenomena of the gluten network, with consequent CO_2_ loss through the bubble wall (Figure 1B, image insert at 48 h) [29].

### 3.2. Rheological Properties of Pizza Dough

The viscoelastic properties of solid foods, combining the properties of a purely viscous fluid and an elastic solid, have frequently been studied by stress-relaxation tests, consisting of applying a compressive force to the specimen until it reaches a final fixed thickness and then observing the force decay during the relaxation stage [18,30,31]. From the relaxation phase graph of pizza dough balls at different leavening times (Appendix A), it is evident that dough samples behave as a viscoelastic material, and it can be deduced that dough has a more elastic component at early times (4 and 8 h), while for longer times it approaches viscous behavior. Nevertheless, the fitting of stress-relaxation curves by using Equation (2) allowed for the determination of several parameters, reported in Table 1, together with the diameter of the disc dough at the end of the test. The compression work corresponds to the work done to deform the dough. The relaxation asymptotic percentage (“*a*”, %) and force decay rate (“*b*”, Ns^−1^) account for the viscoelastic characteristics of the dough. An ideal elastomer that, after a stress, has the ability to recover its original structure instantaneously displays an “*a*” value of 0%. On the contrary, an ideal viscous material (i.e., water) has an “*a*” value equal to 100% [31]. Theoretically, when it comes to viscoelastic solids, the lower the force decay rate, the slower the stress relaxation; conversely, a higher decay rate (b) indicates a fast approach of the relaxation curve toward the asymptotic value [18].

As shown in Table 1, at time 0 h, the deformation of the dough requires the minimum work (6.59 ± 0.41 J), compared to other leavening times, and gives rise to a flat disc with a diameter of 21.50 ± 2.07 cm, the lowest of all (Table 1).

This means that, just after mixing, the dough was weak but at the same time less flowable, thus opposed less resistance to deformation and resulted in a disc with the lowest diameter, relaxing 82.5 ± 0.7% of the applied force with a high decay rate of 0.387 ± 0.014 Ns^−1^ (Table 1). During the first stage of leavening (4 h), the compression work increased almost five times, and the diameter of the formed disc was the highest of all. This was related to the relaxation of internal stresses generated by mixing operations occurring in the early stage of leavening, which change the structure of the dough by making it more compact as well as more extensible and elastic, as confirmed by the significative decrease of relaxation “*a*” (79.7 ± 0.3%) and increase of decay rate “*b*” (0.409 ± 0.007 Ns^−1^) values (Table 1). At this stage, a strong gluten network was formed [32], which could be associated with the high value of compression work. Li et al. [33] reported the study of gluten relaxation, which corresponds to a structure made of a network in which proteins are bound together and a large number of interactions between protein chains leads to an increase in dough strength, as in the case of the dough at 4 h of leavening. At the 8th hour of leavening, the elastic component did not change, as stated by the constancy of the “*a*” parameter (Table 1), but a huge drop in compression work was observed, accompanied by significant reductions in the decay rate “*b*” and in the diameter. With increasing leavening times (16 and 24 h), the decay rate “*b*” and the final diameter after the stress-relaxation test remained almost the same, while the relaxation parameter “*a*” increased again compared to the previous time step (8 h), suggesting valuable changes in the viscous component. Indeed, the time interval between 8 and 24 h comprises most of the growing phase of leavening, in which the fermentative activity of yeasts is responsible for the CO_2_ production by consumption of the reducing sugars, which leads to the filling of the tiny air cells that increase the overall volume of the dough. Thus, the structure becomes porous because of the stretching of the gluten network in which the CO_2_ was retained (Figure 1B, times 16 and 24 h). At 16 h, the work value was higher than at 8 h just for the presence of CO_2_ in the dough, which increased its resistance to deformation by opposing the compression force work (Table 1). As the leavening progresses towards the stationary phase, the dough structure becomes progressively weaker, as shown by the compression work, which significantly reduces with increasing the leavening time, going down to 15.97 ± 1.43 J at 24 h and to 7.65 ± 0.40 J at 48 h (Table 1), due to the further changes in the structure. This suggests a weakening and reorganization of the gluten network and gas bubble wall; thus, the structure becomes less capable of sustaining its own weight, losing part of its elastic behavior. In fact, the accumulation of CO_2_ pushes the gluten mesh to widen so much that at a certain point the gas cells coalesce; simultaneously, the hydrolytic activity of enzymes (endogenous or produced by yeast) degrades starch and weakens the gluten bonds. These two phenomena also explain the loss of elasticity and the increase of the viscous component, as testified by the highest value of the relaxation “*a*” (85.2 ± 0.3%) and the simultaneous lowest value of the decay rate “*b*” (0.283 ± 0.013 N s^−1^) registered at 48 h (Table 1).

The behavior of viscoelastic materials is associated with the structural characteristics of the protein component. Indeed, it is well known that gliadins are associated with dough extensibility; they increased the viscous modulus (G’’) of the dough, while glutenin proteins contribute to the increase in the elastic component described by the elastic modulus (G’) [16,34]. In addition to proteins that play a key role in determining the viscoelastic properties, starch and its damage during leavening also cause interference in the model dough. Interactions between the components of the wheat flour, in particular gluten and starch, which in consequence become visible in the viscoelastic characteristics, can be evaluated by macroscopic examination in dynamic oscillation measurements [35]. Information regarding alterations in the gluten network was obtained from the determination of the elastic modulus (G’) of selected doughs [36,37]. In addition, dynamic rheological measurements have been useful in understanding how starch and gluten interact with each other. In fact, starch not only acts as a filler in the gluten network but also interacts with gluten proteins and is involved in determining the viscoelastic behavior of the dough [38]. During the interaction phase between these compounds, water plays a key role; in fact, the interactions that are formed depend on the ability of starch granules to swell during the gelatinization phase and on the ability of some of these molecules to interact with water itself during their structural changes.

Experimental results for studying the G’ modulus in the dough at different leavening times are shown in Figure 2. During heating from 30 °C to about 60 °C, G’ slightly decreased as the temperature increased, with a linear stretch having an almost parallel slope in all samples except for the dough at 4 h and 48 h of leavening (Figure 2A). This reduction in the elastic modulus was probably due to water being freed from damaged starch in the early stage of heating [39]. As heating continues above 60 °C, two events occur: (1) the gluten proteins denature, providing greater flexibility to the network; (2) the starch starts to gelatinize. In this phase, the G’ curves still decrease along the rising temperature until they reach a minimum value in the range of 70–80 °C (Figure 2A), which corresponds to the overall denaturation of gluten proteins and to the transition of starch from a solid to gel state, resulting in a reduction in viscosity (G’’ modulus; data not shown). The changes in viscosity during gelatinization are well known for starch-containing materials. During the baking phase, starch assumes an important role in rearranging the network [40]. At higher temperatures (>70–80 °C), the trend of G’ curves shows a sudden increase for all selected leavening times. In accordance with Masi [19], the obtained G’ curves are typical for the thermal denaturation of gluten in a temperature sweep test. It has been shown that as the temperature rises, the G’ and G’’ moduli decrease until they reach a critical point, after which there is a reversal of the behavior of these materials, with both moduli increasing [19]. Other studies showed the same behavior of the viscoelastic dough; they also found an abrupt change between 48 and 56 °C, where G’ reached a minimum before starting to increase. This phenomenon has been mainly attributed to starch gelatinization [30,41].

Over the years, several models for the structure of the gluten network have been proposed, often focusing on understanding the viscoelastic properties of the dough. Elongation of the gluten network would result in deformation of protein regions [42]. The functionality of gluten proteins is highly dependent on the specific dough recipe, as water and other typical dough components, such as salt, influence the formation of the gluten network [43]. Yeast and, in particular, its metabolites produced during fermentation also have an impact on the gluten network. Bernklau et al. [43] concluded that the level of free (accessible) SH groups and protein surface hydrophobicity are the main determinants of the gluten protein network during heating.

The G’ curves in the cooling phase could be a useful way to study the gluten network interaction along the leavening time after baking. As shown in Figure 2B, a steep increase in G’ modulus was visible for all samples during the cooling phase from about 90 to 65 °C, which can be attributed to the strengthening of the gluten network through the formation of additional cross-links, such as disulfide bonds, leading to stabilization of the dough structure [14]. At temperatures lower than 65 °C, the G’ curves advanced much more smoothly, with higher slope values for the dough at 4 and 8 h of leavening and an almost flat shape for the unleavened dough (time 0 h) and long leavened samples (16, 24, and 48 h), the latter showing a greater evident reduction in the elastic modulus compared with all other samples (Figure 2B). According to stress relaxation findings (Table 1), this result indicates that, after a long leavening, the gluten network formed after baking has fewer and weaker bonds, and therefore fewer interactions, compared both to the initial dough (time 0 h) and leavened samples for 4 and 8 h. In all cases, the viscous modulus (G”) exhibits the same trend but lower values as the elastic one G’ over the leavening time (data not shown).

### 3.3. Thermal Properties

Wheat flour dough is a heterogeneous system in which thermodynamically incompatible polymers coexist in separate aqueous phases, and thermal property analysis can help unravel the nature and evolution of the interactions between water and these polymers over time [44]. The heat to which the dough is subjected during the thermal analysis mimics what happens during the baking of the pizza. From a physical point of view, during the heating of the dough, it undergoes a significant loss of water, accompanied by the transition of the starch from the crystalline form to the gelatinized one. At the same time, the proteins that make up the gluten network rearrange themselves in new interactions following the gelatinization of the starch. The thermogravimetric profiles of dough at the selected leavening time and corresponding first derivative graphs are shown in Figure 3A,B. After an initial section characterized by a gentle descent of 10% until about 100 °C, TGA curves showed two main steps in weight loss: a first drop from 100 to 250 °C that led to the reduction of another 32% of weight and a further loss of about 28% in the temperature range between 250 °C and 400 °C (Figure 3A). These two steep steps of the thermogram drew two high peaks in the graph of the first derivative, which identified the weight loss rate values associated with two distinct types of events (%/°C) (Figure 3B). It can be assumed that, up to 250 °C, the recorded phenomena concern the movement of water molecules between the two polymeric phases of the dough and between these and the outside. In fact, the heat supplied leads to an increase in the kinetic energy of the molecules, so the free water diffuses more easily according to the gradient from one polymeric phase to another. The result was that the starch tends to hydrate and pass into a gel phase, while the gluten network stretches under the swelling pressure of gelatinized starch, and only a small part of the free water manages to evaporate up to 100 °C, producing a 10% weight loss. At temperatures above 100 °C, water acquires more and more mobility, and not only the free one but also the bound one manages to evaporate; this determines the first important drop in weight of the sample of 32% (Figure 3A), which is characterized by a huge peak in the derivative curve at around 150 °C (Figure 3B). Fessas and Schiraldi [44] have shown that the bound water fraction, which mainly interacts with gluten, was released with a maximum rate reached at 125 °C. Wehrli et al. [45] also found that water tightly linked to gluten was removed at temperatures between 100 and 300 °C and highlighted that, around 250 °C, phenomena of pyrolysis of the matrix material occur, with the release of CO_2_ groups and volatile hydrocarbon species due to the decarboxylation and degradation of proteins. Therefore, the second important weight loss recorded in the TGA curves was due to the thermal decomposition of the dough, which proceeds up to 400 °C and beyond [44]. Therefore, most of the water in the dough, which, taking into account the recipe and the initial moisture of the flour, represents about 50% of the final dough weight, was removed in the temperature range between 100 and 250 °C, and the measurement of the area under the first peak of the derivative curve expresses the extent of the weight loss rate for this water removal. The values of the weight loss rate calculated for the dough at the selected leavening time (Figure 3C) are related to the dough structure.

Experimental evaluations revealed that after 4 and 8 h of leavening, the water removal rate decreased compared to the unleavened sample (0 h), while it increased significantly at 16 h. At the time of 24 h, it remains higher, returning to the initial value at the time of 48 h (Figure 3C). The increase in weight loss rate suggests a more open and weak gluten structure, whereas the decrease could be due to a more compact and stronger gluten network [46]. These results agree with the above considerations regarding the volume during the leavening kinetic (Figure 1), confirming that the more alveolar structure of the dough at 16 h was associated with a more rapid evaporation of the water molecules.

According to the current literature [40,46], DSC analyses of all samples exhibited a single endothermic transition with corresponding temperatures and enthalpies (Table 2). A significative difference between the 16 h leavened sample compared with almost all others, in terms of initial (T_i_) and gelatinization (T_g_) temperatures, as well as in the ∆H value, was found (Table 2).

Interactions between the starch and gluten proteins during the heating phase affect the initial, gelatinization, and final temperatures (T_i_, T_g_, and T_e_), and ∆H. The lower temperatures of the endothermic transition for the 16-h sample (T_i_ = 66.54 ± 0.29 °C, T_g_ = 74.14 ± 0.27 °C) indicate a greater ease of starch hydration, which could be due to a less ordered microstructure with fewer bonds. Hence, during the heating, starch granules start to swell quickly, thus decreasing the content of free water in the surrounding matrix. This happens in a well-bubbled dough following the distension of the gluten network thanks to the expansion of CO_2_ produced by yeast. Further, the barrier effects of gluten on the surface of starch granules are a major obstacle to starch hydration and, thus, gelatinization [40]. The increase in ∆H value as leavening time increased to 16 h (0.975 ± 0.013 J/g) corresponds to a higher gelatinization degree of starch, in turn due to a greater availability of free water. This phenomenon is associated with the destructuring of the dough over the leavening time, which allows water to enter the starch granules more easily as they gel during heating. At a longer leavening, the reduction in ∆H could be attributed to a competition for water between gluten and starch molecules, which become more damaged due to the hydrolytic activities of flour and yeast enzymes [47].

### 3.4. Reducing Sugars, Free Amino Groups, and Digestible Starch in Dough and Pizza

Water-extractable reducing sugars were detected in almost similar amounts in dough and pizza samples along the leavening time interval, with no significant differences for the considered sample at the same time, except at 4 h (Figure 4A). Reducing sugars are expected to increase during leavening as a result of amylolytic activity by both endogenous flour and yeast enzymes, but some of these sugars are used as fermentable substrates by yeasts, so the measured values correspond to residual sugars taking into account the yeast metabolic activity [48]. Thus, an increasing trend of reducing sugar levels was registered up to 16 h of leavening, followed by a reduction at 24 and 48 h with significant differences compared to 0 h (unleavened) and to 48 h samples for the dough and pizza samples (Figure 4A). Meerts et al. [49] reported that the highest yeast microbial activity occurs up to 6 h after leavening, after which it settles. This agrees with the results of this study, which show an increase in sugars up to 4 h and then remain almost constant. The presence of amino acids and peptides in aqueous extracts of dough and pizza samples was evaluated indirectly by the determination of free primary amino groups. The results, expressed as g of leucine equivalents/100 g of sample on a dry weight basis, showed comparable values in the first 8 h in the case of dough samples, with a slight increase in the following hours of leavening (Figure 4B). 

Several studies have explained this trend as a result of the proteolytic activity of the flour [48]. Probably, this increase was due to alterations in the gluten network due to the metabolic activity of microorganisms during leavening. Nielsen et al. [50] suggested that the OPA method can be used to determine the degree of hydrolysis in hydrolyzed proteins in foods. In this sense, it is possible to hypothesize that free peptides show an upward trend during the leavening of dough as a result of endogenous proteolytic activity and lactic acid bacteria naturally present in flour [51]. For pizza samples, however, a slight increasing tendency was observed from 3.42 ± 0.27 g/100 g_dw_ of sample at time 0 h to 4.64 ± 0.18 g/100 g_dw_ after 48 h of leavening. In each case, the measured values were lower with respect to dough, indicating that baking results in matrix modification because, at high temperatures, amino acids and peptides tend to react with sugars according to the Maillard reaction. In addition, proteins aggregate more stably, thus affecting the efficiency of amino acid release from ends by proteases.

For nutritional purposes, starch has been classified according to the rate of digestion into rapidly digestible starch (RDS, 20 min), slowly digestible starch (SDS, 120 min), and resistant starch (RS, not digested after 240 min) [52]. The rate and degree of digestion are widely dependent on the structure [53]. RDS is rapidly and completely digested in the small intestine and is associated with a more rapid elevation of postprandial plasma glucose, whereas SDS is more slowly digested in the small intestine and is generally the most desirable form of dietary starch [54].

Results for *in vitro* starch digestion of dough and pizza base samples are reported in Table 3. The results obtained from the dough samples showed a similar amount of RDS to the amount of SDS after 120 min of enzymatic digestion. Specifically, in dough, the RDS value was constant between 0 and 8 h of leavening, and in subsequent hours it decreased following the metabolism of the yeasts; no significant difference in SDS values was found for all the samples, while RS was very high, ranging between 36.89 ± 0.33 at 4 h of leavening and 47.99 ± 0.43 at 48 h, due to the crystalline physical state of starch (Table 3A).

In wood oven-baked pizza base samples, RDS increased up to 4–9 times compared to the dough due to the starch gelatinization that occurs during baking, which makes it much more susceptible to α-amylase hydrolysis, and therefore, the share of RS was practically negligible in pizza bases.

Moreover, a significant difference was observed between time 0 and 4 h of leavening, with an increase in RDS and a decrease in SDS. From 8 h onward, there was a clear increase in RDS at the expense of SDS. These results strengthen the finding that pizza is a high glycemic index (GI) food; in fact, after baking, all the starch becomes rapidly digestible from the eighth hour at long leavening time, while the SDS value tends to almost zero at long leavening time, in contrast to the dough samples, where the SDS levels were found to be almost similar to the RDS values (Table 3). The rapidly available glucose index, named RAG, has been described as a predictor of the potential glycemic response derived from the ingestion of a food and is nothing more than the RDS [24]. In fact, increased RDS and decreased SDS are also associated with increased glycemic response [55,56], and the importance of the relative content of the two starch fractions, RDS and SDS, was also evident in other studies conducted by Ells et al. [57]. Indeed, they showed that postprandial glucose and insulin varied significantly after consumption of foods with higher or lower RDS and SDS. In this sense, the SDRI parameter was the most useful tool in predicting the GI of pizza samples with different water and TS contents. From the results shown in Table 3B, it can be stated that in wood oven-baked pizza samples, starting from 8 h of leavening, an increase in starch digestibility occurs, which correlates with an increase in the glycemic index after gastric digestion, as predicted by RAG values (Table 3B). The resistant starch (RS) levels in the dough settled around 40 percent, decreasing from 44.07 ± 1.75 to 36.89 ± 0.33% between 0 and 4 h, and then slightly increasing for the following leavening time to 47.99 ± 0.43% at 48 h (Table 3A). In the case of the pizza samples, however, the results showed an almost negligible amount of resistant starch, and this is a further confirmation that pizza is a high glycemic index food.

### 3.5. Acrylamide Levels in Pizza Base Leavened for Different Times

Acrylamide (AA) is formed mainly in the Maillard reaction from asparagine and carbonyl sources, and this reaction occurs mostly above 120 °C [58,59]. Several factors influence acrylamide formation, such as reducing sugars, asparagine, fiber contents, damaged starch, fermentation time, and moisture of the sample. As a starchy product baked at a high temperature in a wood oven, Neapolitan pizza is considered a risky food for its AA content. There are no studies in the literature reporting the AA content of Neapolitan pizza baked in a wood oven. An in-depth study for the reduction of AA levels in pizza bases using asparaginase enzyme has been previously conducted [9]. In that study, it was reported that the average AA content in a wood oven-baked pizza base was around 2000 µg/kg_dw_. To evaluate the impact of leavening time on the formation of this toxic substance, wood oven-baked pizza base samples prepared at different leavening times (0, 4, 8, 16, 24 and 48 h) were tested for AA content.

From the experimental data, the levels of AA varied from 634 to 1703 µg/kg_dw_, with no significative differences between 4 and 48 h of leavening (Figure 5). These results can be explained by considering the levels of both reducing sugars and free amines, taken as references for free amino acids (including asparagine), during leavening (Figure 4), the two important precursors in the formation of AA. In fact, between 0 and 4 h, the reducing sugars in the dough increased significantly by one point, after which they remained within a range of 3.5–3.8 g/100 g_dw_ up to 24 h, returning to the starting value after 48 h of leavening. The levels of free amino groups, on the other hand, fluctuate around a value of 4.3 g/100 g_dw_ for the entire leavening period, with a significant increase to around 5.6 g/100 g_dw_ at 48 h. Therefore, given that the most important variation of reducing sugars was recorded between 0 and 4 h, the formation of AA also increased in this time interval, and the values did not undergo significant differences for the rest of the leavening.

Unlike what was expected, despite the collapse of reducing sugars at 48 h, no lowering of the AA value was observed, probably due to a simultaneous increase in free asparagine in the dough, which keeps the value high (Figure 4).

Finally, it can be hypothesized that in the unleavened product, the AA value was significantly lower due to issues related to baking. As soon as the dough was placed in the oven, water evaporated very quickly from the surface layers, resulting in a much lower water content than at the core. As the water content decreases in the external part of the pizza base, the temperature can exceed 100 °C, which supports reactions such as caramelization, carbonization, and the Maillard reaction, responsible for the brown coloration [60,61]. Therefore, unleavened dough was less malleable and stretchable, resulting in a more compact pizza disc, and the structure of the dough matrix did not favor the migration of the water during baking. Since water is an obstacle for the formation of AA, a smaller amount of AA was produced in the unleavened pizza due to the persistence of a certain amount of hydration during the baking process in the wood oven. On the other hand, at 48 h of leavening, with the same reducing sugar values as at time 0, the AA level reached was higher because the dough allows for a rapid removal of water during baking, which triggers the Maillard reaction, which leads to the formation of AA.

Beyond all these considerations, it is important to underline that, in any case, the average AA value determined for the wood oven-baked Neapolitan pizza base TSG of ~1700 µg/kg_dw_ was well below the alert threshold set by EFSA for daily intake of AA (0.170 mg/kg of body weight).

### 3.6. Morphology of Dough and Pizza Base during Leavening

SEM analyses allowed for the description of the morphology of dough and pizza samples over leavening time (Figure 6). At 0 h, a disordered porous structure was visible in the dough sample; the holes can be attributed to the presence of air incorporated during the kneading of the ingredients. After 4 h, the cavities disappeared, and the structure was much more compact, with the wheat starch granules in the foreground being spherical or oval in shape [62]. At 8 h, the gluten network, developed under the CO_2_ pressure, started to cover the starch granules; between 16 and 24 h of leavening, the protein filaments characteristic of a flaking network appeared, and the gluten structure was broken and discontinuous. Finally, after 48 h, the network collapsed completely and the structure returned to being compact (Figure 6A). In fact, increasing the leavening time of pizza dough up to 48 h can result in increased starch hydrolysis that leads to the destructuring of the gluten, both for the loss of the physical support of starch granules and the weakening of the protein network and wall barrier properties to gas permeation.

According to our result, Yan et al. [63] reported that leavened dough with starter for 4 h showed conspicuous starch granules and a firm and dense gluten network. The starch granules were embedded in the gluten and combined rigidly in the first few hours of leavening. It has also been demonstrated that, as the leavening time increases, depolymerization of macromolecule proteins and the formation of gluten proteins with a fibrous structure and greater continuity are observed [64]. Moreover, other studies have shown the interaction between gluten and starch matrices, as evidenced by the smooth and uniform character of the contact interface [21,65]. Previous studies reported that at high leavening times, the dough loses its ability to hold gas, resulting in products with poor elasticity [65], both due to the pH lowering that leads to the degradation of polymers, promoting the opening of the gluten network, and the disruption of intermolecular disulfide bonds between glutenin and gliadin, which makes starch grains more exposed to the action of hydrolytic enzymes [66]. SEM observation of wood oven-baked pizza base samples highlighted that, even after baking, the unleavened sample (0 h) presented a porous structure in which protein filaments and starch granules were evident (Figure 6B).

After 4 and 8 h of leavening, a disappearance of the holes was found with the obvious formation of protein-starch bonds, confirming the results obtained with mechanical-dynamic rheological tests. Indeed, starch not only acts as a filler in the gluten network but also interacts with gluten proteins. As the leavening time increased, the proteins formed a “sheet” layer that covered the starch granules, which was most evident at time 16 h. A “stretching” of the gluten network was observed at 24 and 48 h of leavening, with starch granules more exposed but swollen and burst because of gelatinization during the heating process (Figure 6B). Lindsay and Skerritt [67] also showed in earlier studies that gliadins were uniformly dispersed within gluten filaments throughout the dough, consistent with a “space-filling” role.

## 4. Conclusions

For the first time in this work, a very detailed study of pizza dough during prolonged leavening has been undertaken, evaluating both physical and structural as well as rheological and biochemical parameters. The results showed that the volume of the dough doubled after 16 h and tripled in 24 h, reaching a constant value around 30 h, which did not vary up to 48 h. In this period of time, the rheological characteristics of the dough change: at 4 h, the maximum compression work value was recorded (29.96 ± 2.52 J), the lowest value of the relaxation percentage “*a*” (79.7 ± 0.3%), and the highest decay rate “*b*” (0.409 ± 0.007 Ns^−1^), resulting in the maximum diameter of the formed pizza disc. In this stage, the gluten network was very strong and showed a high value of elastic modulus (G’), whose reduction during leavening accounts for the loss of elasticity to the benefit of the increase in the viscose component. At longer leavening times, the work for compressing the dough decreased due to the weakening of the gluten mesh and the interactions with starch. In fact, the diffusion in the dough of the CO_2_ produced by the yeasts has the effect of widening the meshes of the gluten network. This was also confirmed by the images obtained with SEM, which highlighted the presence of stretched protein filaments, starting from 8 h of leavening, which trap the starch granules. The increased exposure of starch promoted its damage by α-amylase, while proteases may have easier access to proteins. Damage to the starch at higher leavening times led to a greater propensity for gelatinization during baking; in fact, the free water contained in the dough, as the gluten network relaxed, was more easily absorbed by the damaged starch that hydrates. Thermal analyses at TGA and DSC confirmed that doughs with longer leavening retained more bound water and showed an increase in the temperature range of the solid-gel transition and in the associated ΔH therein.

Another consequence of the hydrolytic enzyme’s action was the release of reducing sugars and free amino compounds, such as amino acids and peptides, which increased during leavening and persisted even after baking. Nonetheless, apart from an initial increase from 0 to 4 h, the acrylamide levels in the pizza base remained almost constant at the other times from 4 h to 48 h, indicating little influence of the leavening time on acrylamide formation.

Along with the increase in reducing sugars, the percentage of rapidly digestible starch (RDS) decreased in the dough, precisely because it was degraded and partly consumed by the yeasts. With baking, RDS levels became high due to gelatinization, which transformed resistant starch into digestible starch. Furthermore, the longer the leavening time, the higher the percentage of RDS, suggesting that pizza bases leavened for a long time were more digestible. Unfortunately, this also leads to an important value of RAG (rapidly available glucose index), which is predictive of the high glycemic index of the pizza.

Based on all the results obtained in this study, from a practical point of view, it can be concluded that a long leavening improves the extensibility of the pizza disc, facilitating the action of the “*pizzaiuolo*” (pizza maker), makes the pizza more digestible, and all in all, does not influence the formation of acrylamide, whose value falls within the safety limits for the daily intake established by EFSA.

## Figures and Tables

**Figure 1 foods-12-01407-f001:**
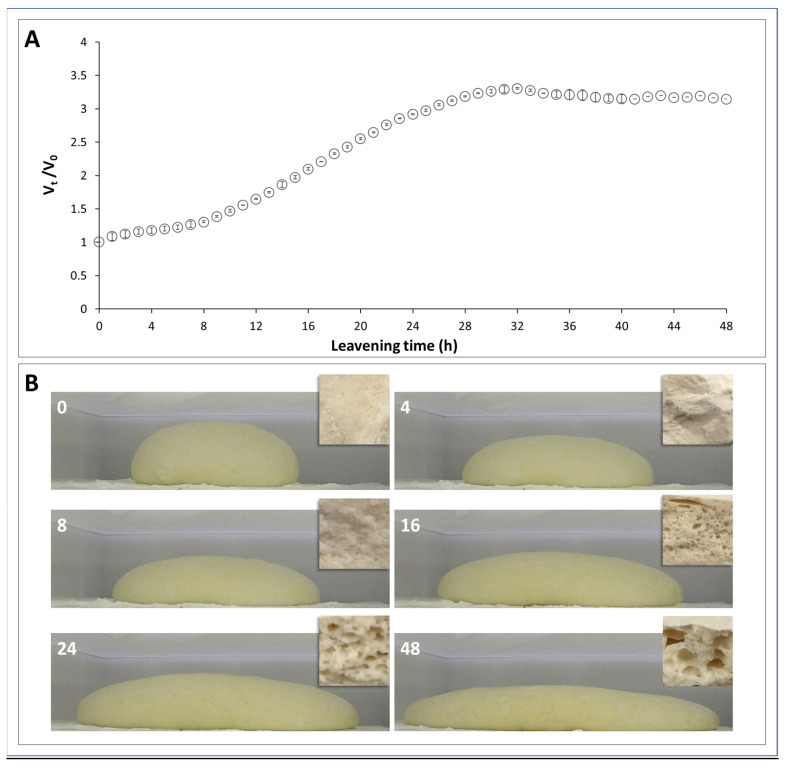
Leavening kinetics of pizza dough. (**A**) Variation of volume index (V_t_/V_0_) over time (see also Appendix A); (**B**) images taken at 0, 4, 8, 16, 24 and 48 h. The images inserted at the top right of each photo refer to sections of dough at the corresponding leavening time.

**Figure 2 foods-12-01407-f002:**
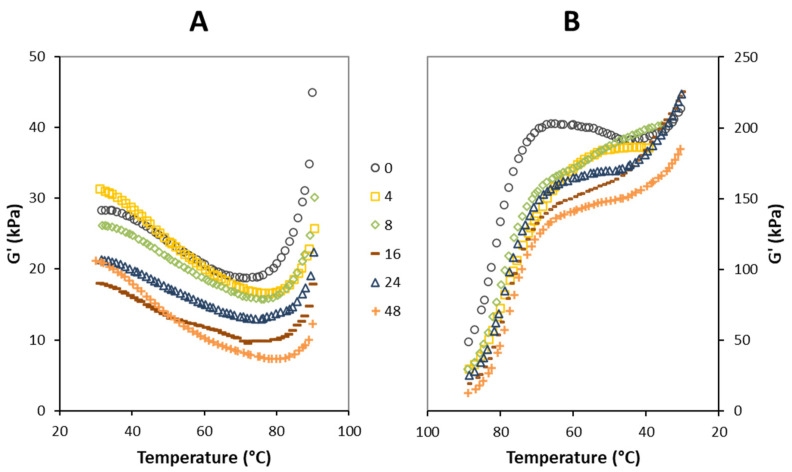
Elastic modulus (G’) of pizza dough at the selected leavening time during the heating (**A**) and cooling (**B**) phases.

**Figure 3 foods-12-01407-f003:**
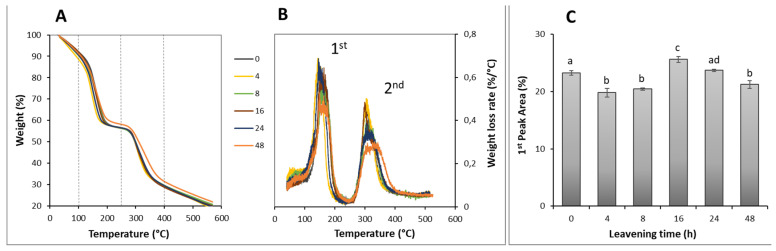
Thermogravimetric analysis (TGA) of pizza dough at increasing leavening times. (**A**) TGA curves as functions of temperature. (**B**) first derivate of TGA curves. (**C**) First peak area from the derivate curves. Different letters indicate samples significance calculated by an ANOVA statistical test with Post Hoc Tukey (*p* < 0.05).

**Figure 4 foods-12-01407-f004:**
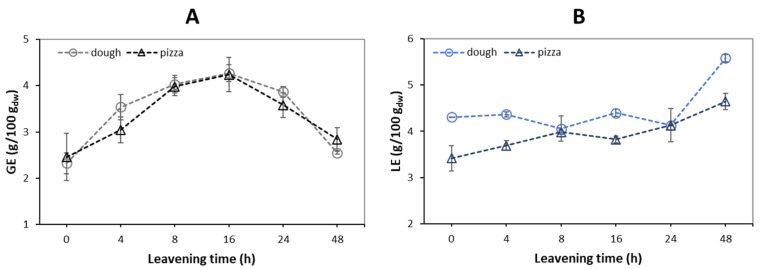
Reducing sugars (**A**) and free amino groups (**B**) of the dough and wood oven-baked pizza base at a selected leavening time. Values are mean ± SD (*n* = 4). GE, glucose equivalent reducing sugars; LE, leucin equivalent free amino groups.

**Figure 5 foods-12-01407-f005:**
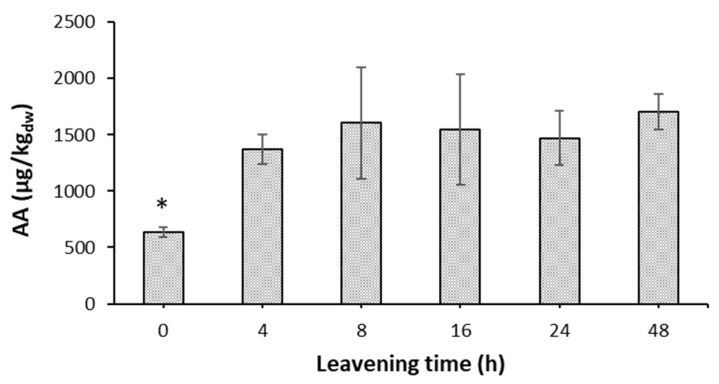
The AA content of a wood oven-baked pizza base leavened for 0, 4, 8, 16, 24, and 48 h. Values are mean ± SD (*n* = 3). *, significantly different by ANOVA statistical test with Post Hoc Tukey (*p* < 0.05).

**Figure 6 foods-12-01407-f006:**
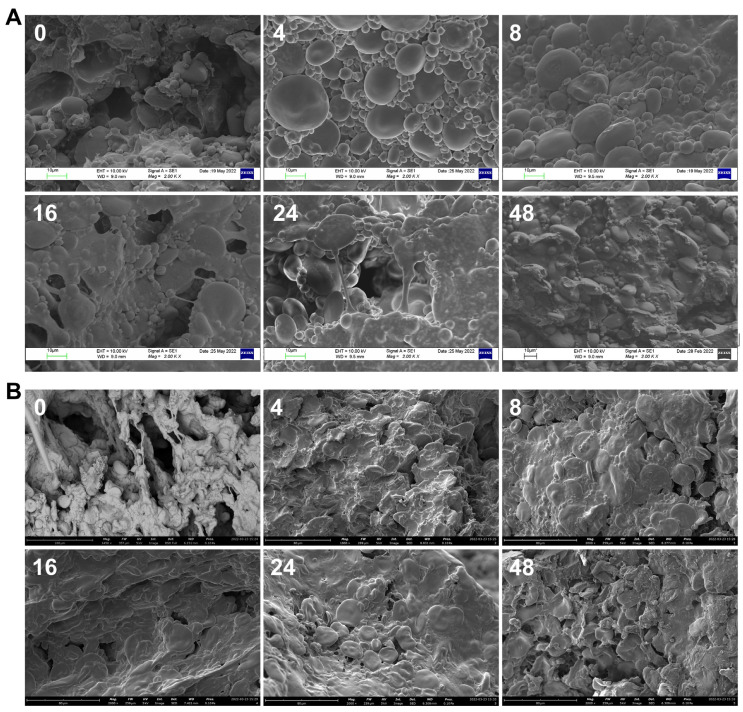
SEM images of dough (**A**) and a wood oven-baked pizza base (**B**) at a selected leavening time. Magnification: 2000×.

**Table 1 foods-12-01407-t001:** Compression work (J), relaxation percentage (*a*), decay rate (*b*), and diameter reached by pizza dough after stress-relaxation test at selected leavening time.

Leavening Time (h)	Compression W (J)	Relaxation *a* (%)	Decay Rate *b* (N s^−1^)	Diameter (cm)
0	6.59 ± 0.41 ^a^	82.5 ± 0.7 ^a^	0.387 ± 0.014 ^a^	21.50 ± 2.07 ^a^
4	29.96 ± 2.52 ^b^	79.7 ± 0.3 ^b^	0.409 ± 0.007 ^b^	29.25 ± 0.53 ^b^
8	16.16 ± 0.69 ^c^	78.9 ± 0.4 ^b^	0.342 ± 0.008 ^c^	26.06 ± 1.05 ^c^
16	19.72 ± 1.10 ^d^	80.8 ± 0.3 ^c^	0.359 ± 0.015 ^c^	27.50 ± 0.46 ^cd^
24	15.97 ± 1.43 ^c^	80.9 ± 1.0 ^c^	0.347 ± 0.010 ^c^	28.37 ± 0.44 ^bd^
48	7.65 ± 0.40 ^a^	85.2 ± 0.3 ^d^	0.283 ± 0.013 ^d^	28.06 ± 0.42 ^bd^

Values are mean ± SD (*n* = 8). Different letters indicate samples significance calculated by an ANOVA statistical test with Post Hoc Tukey (*p* < 0.05).

**Table 2 foods-12-01407-t002:** DSC analysis of pizza dough at selected leavening times, expressed as initial gelatinization, ending temperature (T_i_, T_g_, and T_e_), and enthalpy variation (∆H).

Leavening Time (h)	T_i_ (°C)	T_g_ (°C)	T_e_ (°C)	∆H (J/g)
**0**	68.17 ± 0.01 ^a^	75.40 ± 0.01 ^a^	85.15 ± 0.33 ^a^	0.613 ± 0.015 ^a^
**4**	68.09 ± 0.47 ^a^	75.12 ± 0.30 ^a^	86.40 ± 0.42 ^a^	0.709 ± 0.056 ^b^
**8**	68.08 ± 0.65 ^a^	75.53 ± 0.03 ^a^	86.70 ± 1.12 ^a^	0.805 ± 0.045 ^b^
**16**	66.54 ± 0.29 ^b^	74.14 ± 0.27 ^b^	87.46 ± 0.42 ^a^	0.975 ± 0.013 ^c^
**24**	68.29 ± 0.27 ^a^	75.39 ± 0.22 ^a^	88.00 ± 0.67 ^a^	0.917 ± 0.036 ^d^
**48**	67.65 ± 0.63 ^ab^	74.36 ± 0.08 ^b^	86.94 ± 1.68 ^a^	0.730 ± 0.090 ^b^

Values are mean ± SD (*n* = 3). Different letters indicate samples significance calculated by an ANOVA statistical test with Post Hoc Tukey (*p* < 0.05).

**Table 3 foods-12-01407-t003:** *In vitro* digestible starches of dough (**A**) and wood oven-baked pizza base (**B**) at selected leavening times.

	Leavening Time (h)	RDS	SDS	RS	TS	SDRI	RAG
		(%)					
**A**							
** *dough* **	0	12.15 ± 0.96 ^a^	12.48 ± 0.70 ^a^	44.07 ± 1.75 ^a^	68.71 ± 3.42 ^a^	-	-
	4	11.81 ± 0.07 ^a^	12.06 ± 0.54 ^a^	36.89 ± 0.33 ^b^	60.76 ± 0.29 ^a^	-	-
	8	13.29 ± 0.54 ^a^	11.43 ± 1.44 ^a^	38.59 ± 1.05 ^b^	63.32 ± 3.04 ^a^	-	-
	16	9.76 ± 0.36 ^b^	12.22 ± 2.02 ^a^	38.83 ± 0.62 ^b^	60.81 ± 2.28 ^a^	-	-
	24	7.25 ± 0.12 ^c^	12.12 ± 1.27 ^a^	46.78 ± 2.93 ^a^	66.16 ± 4.33 ^a^	-	-
	48	9.25 ± 0.39 ^b^	8.72 ± 0.59 ^a^	47.99 ± 0.43 ^a^	65.96 ± 0.23 ^a^	-	-
**B**							
** *pizza base* **	0	46.06 ± 3.01 ^a^	26.08 ± 5.87 ^a^	0.58 ± 0.05 ^a^	72.72 ± 2.82 ^a^	63.46 ± 6.60 ^a^	51.18 ± 3.44 ^a^
	4	53.25 ± 1.07 ^ab^	11.53 ± 5.79 ^b^	0.50 ± 0.01 ^a^	65.28 ± 4.73 ^a^	81.84 ± 7.57 ^b^	59.17 ± 1.19 ^ab^
	8	65.65 ± 0.75 ^c^	1.62 ± 0.76 ^c^	0.48 ± 0.03 ^a^	67.75 ± 1.55 ^a^	96.91 ± 1.11 ^c^	72.95 ± 0.83 ^c^
	16	67.15 ± 0.87 ^c^	1.63 ± 0.87 ^c^	0.44 ± 0.02 ^a^	69.23 ± 0.02 ^a^	97.00 ± 1.28 ^c^	74.61 ± 0.97 ^c^
	24	64.58 ± 3.93 ^c^	1.67 ± 1.13 ^c^	0.45 ± 0.07 ^a^	66.71 ± 2.87 ^a^	96.78 ± 1.73 ^c^	71.76 ± 4.37 ^c^
	48	59.02 ± 2.19 ^bc^	3.22 ± 0.76 ^c^	0.49 ± 0.02 ^a^	62.73 ± 1.45 ^a^	94.07 ± 1.32 ^c^	65.58 ± 2.44 ^bc^

RDS, after 20 min; SDS, after 120 min; RS, after 240 min. Dash (-), not calculated. Values are mean ± SD (*n* = 3). Different letters indicate samples significance calculated by an ANOVA statistical test with Post Hoc Tukey (*p* < 0.05).

## Data Availability

The data are available from the corresponding author.

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
