# Peer review of "Study of Physico-Chemical Properties of Dough and Wood Oven-Baked Pizza Base: The Effect of Leavening Time"

_foods, 2023, doi:10.3390/foods12071407_

Round 1

Reviewer 1 Report

The manuscript is scientifically sound and meets the journal's expectations but needs minor revision. 

1.               There are numerous typos (75). Please check throughout the manuscript. Please make a minor revision of the manuscript that would reorganize and improve its content. Editing for grammar is required.

2.               The material and methods section needs to be concise as methods are presented in long paragraphs and letters should be running; required revision.

3.               The conclusion part require revision as results are not explained well.

4.       Different letters indicate samples significance calculated by ANOVA statistical test with Post Hoc Tukey need to be verify again. 

Author Response

Manuscript ID: foods-2262376

Response to Reviewer 1 Comments

General comment: The manuscript is scientifically sound and meets the journal's expectations but needs minor revision.

Response comment: Many thanks for these kind words. We carefully read all your suggestions and modified the text accordingly. We answered your requests point by point and the modifications in the text are highlighted by review mode.

Q1: There are numerous typos (75). Please check throughout the manuscript. Please make a minor revision of the manuscript that would reorganize and improve its content. Editing for grammar is required.

R1: Thank you for the suggestions. We have checked for grammar or spelling errors and corrected.

Q2: The material and methods section needs to be concise as methods are presented in long paragraphs and letters should be running; required revision.

R2: Thank you for the suggestion. We have shortened the long paragraph # 2.8 and put the letters “a” and “b” in italic style. We hope that this was what you meant when you said “letters should be running”.

Q3: The conclusion part require revision as results are not explained well.

R3: Thank you for the suggestion, we have rewritten the conclusion paragraph.

Q4: Different letters indicate samples significance calculated by ANOVA statistical test with Post Hoc Tukey need to be verify again.

R4: Thank you for the suggestion, we have verified the statistical analysis and corrected in the tables.

Reviewer 2 Report

The manuscript is written well and the theme is good but still, there are many minor changes required to improve the manuscript. Here are few general changes recommended

1.      The title of the manuscript should be revised

2.      The article as well as the title have been drafted without a clear rationale

3.      Abstract needs a lot of attention and should cover theme of whole manuscript

4.      The abstract should focus on the findings of the manuscript.

5.      A conclusive line should also be added at the end of abstract.

6.      Spelling errors were observed in some instants

7.      Grammatical errors in several places

8.      Summarize updated recent research related to the topic

9.      Highlight gaps in current understanding or conflicts in current knowledge

10.  Establish the originality of the research aims by demonstrating the need for investigations in the topic area

11.  Give a clear idea of the target readership, why the research was carried out and the novelty and topicality of the manuscript

12.  There is a lack of discussion in the manuscript. Discuss with their mechanism. How and why changes occurred throughout the manuscript

13.  Graphics of Figures should be improved.

14.  Elaborate the clear results. Recheck it

15.  Discussion should be improved by discussing the different trends.

16.  Only use scientific words throughout the manuscript

17.  The conclusion should be revised for improving the reader's understanding

18.  All the references should be aligned with the same format and according to journal criteria

Author Response

Manuscript ID: foods-2262376

Response to Reviewer 2 Comments

General comment: The manuscript is written well and the theme is good but still, there are many minor changes required to improve the manuscript. Here are few general changes recommended.

Response comment: Many thanks for these kind words. We carefully read all your suggestions and modified the text accordingly. We answered your requests and the modifications in the text are highlighted by review mode. Thanks to your comments the overall quality of the paper was improved.

Q1: The title of the manuscript should be revised

R1: Thank you for the suggestions. We have changed the title in :”Study of physico-chemical properties of dough and wood oven baked pizza base: effect of leavening time”

Q2-5:.

  1. The article as well as the title have been drafted without a clear rationale
  2. Abstract needs a lot of attention and should cover theme of whole manuscript
  3. The abstract should focus on the findings of the manuscript.
  4. A conclusive line should also be added at the end of abstract.

R2-5: Thank you for the suggestion. We have deeply revised the abstract following your indications.

Q6-7:

  1. Spelling errors were observed in some instants
  2. Grammatical errors in several places

R6-7: Thank you for the suggestions. We have checked for grammar or spelling errors and corrected.

Q8-11:

  1. Summarize updated recent research related to the topic
  2. Highlight gaps in current understanding or conflicts in current knowledge
  3. Establish the originality of the research aims by demonstrating the need for investigations in the topic area
  4. Give a clear idea of the target readership, why the research was carried out and the novelty and topicality of the manuscript

R8-11: Thank you for the suggestion, we have followed your indications and implemented the introduction by adding a new paragraph by reporting with recent research papers and evidenced the gap in the literature about Neapolitan pizza TSG.

Q12-16:

  1. There is a lack of discussion in the manuscript. Discuss with their mechanism. How and why changes occurred throughout the manuscript
  2. Graphics of Figures should be improved.
  3. Elaborate the clear results. Recheck it
  4. Discussion should be improved by discussing the different trends.
  5. Only use scientific words throughout the manuscript

R12-16: Thank you for the suggestion, we have implemented the comments and discussion of results throughout the manuscript and checked for the scientific language. We have also reformatted the table 3. We only don’t understand the query of the point n. 13: in our opinion the resolution of figures is high, the numbers and letters on graphs are all readable, the colors are distinguishable, the graph indicators are different. If it is necessary, we can upload the pictures of the figures as jpg file at 300 dpi resolution.

Q17: The conclusion should be revised for improving the reader's understanding

R17: Thank you for the suggestion, we have re-written the conclusion paragraph.

Q18: All the references should be aligned with the same format and according to journal criteria

R18: Thank you for the suggestion, we have corrected the references and added the DOI when available

Reviewer 3 Report

I read with interest the manuscript ID: foods-2262376 entitled “Effect of leavening time on rheological, textural, thermal and chemical properties of dough and wood oven baked pizza base”. The authors investigate how different leavening times affect the structure of pizza dough during fermentation and how rheological and biochemical parameters may be related to the starch digestibility of the baked product.

The manuscript is well written and presents well designed experiment with adequately statistical approach.

Material section provide details on methods used. The authors made a complex evaluation by using different laborious methods of analysis, evaluating both physical and structural as well as rheological and biochemical parameters.

The results and the discussions were detailed and were lead very well with board discussion with literature.

Conclusions are properly drawn.

Provided literature is relevant to the research.

This very detailed study on dough and pizza samples provides comprehensive and detailed information for researcher and consumers.

Congratulations!

Minor comments:

Line 90: 60 s instead of 60 seconds

Line 286: Eq. 2 instead of eq. 2

Line 287-288: please verify the sentence: The compression work corresponds to the work ...

Author Response

Manuscript ID: foods-2262376

Response to Reviewer 3 Comments

General comment: I read with interest the manuscript ID: foods-2262376 entitled “Effect of leavening time on rheological, textural, thermal and chemical properties of dough and wood oven baked pizza base”. The authors investigate how different leavening times affect the structure of pizza dough during fermentation and how rheological and biochemical parameters may be related to the starch digestibility of the baked product.

The manuscript is well written and presents well designed experiment with adequately statistical approach.

Material section provide details on methods used. The authors made a complex evaluation by using different laborious methods of analysis, evaluating both physical and structural as well as rheological and biochemical parameters.

The results and the discussions were detailed and were lead very well with board discussion with literature.

Conclusions are properly drawn.

Provided literature is relevant to the research.

This very detailed study on dough and pizza samples provides comprehensive and detailed information for researcher and consumers.

Congratulations!

Response: First of all we would like to address the reviewer with words of gratitude for the compliments on the appreciation of our work. All the highlighted query were corrected in the text.

Q1: Line 90: 60 s instead of 60 seconds

R1: Thank you for the suggestion, we have corrected in line 174

Q2: Line 286: Eq. 2 instead of eq. 2

R2: Thank you for the suggestion, we have corrected in line 404

Q3: Line 287-288: please verify the sentence: The compression work corresponds to the work ...

R3: Thank you for the suggestion, we corrected in lines 405-406
